# Effect of Bladder Injection of OnabotulinumtoxinA on the Central Expression of Genes Associated with the Control of the Lower Urinary Tract: A Study in Normal Rats

**DOI:** 10.3390/ijms232214419

**Published:** 2022-11-20

**Authors:** Soultana Markopoulou, Lina Vardouli, Fotios Dimitriadis, Dimitra Psalla, Alexandros Lambropoulos, Apostolos Apostolidis

**Affiliations:** 1Department of Pharmacology, School of Medicine, Aristotle University of Thessaloniki, 541 24 Thessaloniki, Greece; 21st Department of Urology, Aristotle University of Thessaloniki, ‘G. Gennimatas’ General Hospital, 541 24 Thessaloniki, Greece; 3Veterinary School, Aristotle University of Thessaloniki, 541 24 Thessaloniki, Greece; 41st Department of Obstetrics & Gynaecology, Molecular Biology Lab, Aristotle University of Thessaloniki, 541 24 Thessaloniki, Greece; 52nd Department of Urology, Aristotle University of Thessaloniki, ‘Papageorgiou’ General Hospital, Ring Road, Nea Efkarpia, 56 403 Thessaloniki, Greece

**Keywords:** onabotulinumtoxinA, bladder, detrusor, central nervous system, spinal cord

## Abstract

To investigate a possible central mechanism of action of Botulinum toxin A (BoNT/A) following injection in the bladder, complementary to the acknowledged peripheral bladder effect, we studied changes in the expression of neuropeptides and receptors involved in lower urinary tract function in the spinal cord (SC) and dorsal root ganglia (DRG) of normal rats following BoNT/A bladder injection. Thirty-six Sprague-Dawley rats, divided into three groups of n = 12, received bladder injections of 2U or 5U OnabotulinumtoxinA (BOTOX^®^), or saline. Six animals from each group were sacrificed on days 7 and 14. Expression of Tachykinin 1 (Tac1), capsaicin receptor (TRPV1), neuropeptide Y (NPY), proenkephalin (PENK) and muscarinic receptors M1, M2, M3, was evaluated in the bladder, L6-S1 DRG, and SC segments using real-time PCR and Western blotting. Real-time PCR revealed increased expression of NPY in all tissues except for SC, and increased TRPV1 and PENK expression in DRG and SC, whereas expression of Tac1, M1 and M2 was decreased. Less significant changes were noted in protein levels. These findings suggest that bladder injections of OnabotulinumtoxinA may be followed by changes in the expression of sensory, sympathetic and cholinergic bladder function regulators at the DRG/SC level.

## 1. Introduction

Since its first successful application of alleviating intractable urinary incontinence in patients with spinal cord injury [1], the clinical efficacy of Botulinum neurotoxin type A (BoNT/A) has been proven by several randomized controlled studies. BOTOX^®^ (OnabotulinumtoxinA) has received approval for treating refractory neurogenic and non-neurogenic overactive bladders, but the exact mechanism of its action in the human overactive bladder has only partly been elucidated. Current knowledge supports a mechanism of action in the bladder different to the one described in skeletal muscle, which is based on the inhibition of acetylcholine (ACh) release in the neuromuscular junction, leading to a temporary reduction in the motor nerve activity and to a muscle state of “flaccid paralysis” [2]. Neuronal activity gradually returns via nerve sprouting from the parental axon terminal which subsides when the main axon terminal returns to its normal physiological state [2].

Such mechanisms have yet to be fully confirmed in the bladder. In axon terminals of the human detrusor muscle, both the receptor for BoNT/A (Synaptic Vesicle protein 2—SV2 receptor), and the substrate upon which the toxin acts, protein SNAP25 (Synaptosomal Associated Protein), have been detected in abundance [3]. In laboratory animals, bladder BoNT/A injection caused a temporary reduction in ACh release from detrusor strips [4], while in the human detrusor, post-BoNT/A reductions in muscarinic receptors were found [5]. These findings, in tandem with the reduction in detrusor contractility, suggest a BoNT/A effect on the detrusor [6]. On the other hand, in vitro experiments in animal models have failed to demonstrate a detrusor effect [7]. In the human overactive bladder, expression of the gap junction protein Cx43 and the interstitial cell protein c-kit remained unchanged with BoNT/A treatment [8], suggesting the absence of action on autonomous endogenous contractions of the bladder [9]. Furthermore, in contrast with the gradual development of nerve sprouts from the affected axon terminals following BoNT/A injection in skeletal muscle, BoNT/A injection in the detrusor muscle does not appear to induce significant nerve sprouting [2].

These diverse mechanisms of action have been incorporated into theories [6] which propose that injection of BoNT/A in the overactive bladder has a complex inhibitory effect on urothelial and suburothelial receptors, as well as on neuropeptides and neurotransmitters involved in the afferent pathways which are important factors/inducers of inherent or spinal reflexes thought to participate in the pathophysiological mechanism of the overactive bladder. Interestingly, earlier animal studies had described changes in neurotransmitter and growth factor gene expression in the spinal cord following BoNT/A injection in peripheral muscles [10,11]. The explanation for this central effect remains to be elucidated. Wiegand et al. detected radioactivity in motoneurons after application of radioiodine-labeled botulinum toxin; the authors explained this finding by proposing a retrograde transport of the toxin [12]. Moreover, they hypothesized that either the intact molecule, or only a part of it with the radiolabel, is retrogradely transported.

This hypothesis has been partly tested in a recent experiment in normal rats which provided evidence for BoNT/A’s transport to the central nervous system (CNS) following bladder intradetrusor injection of the purified toxin [13]. Long-distance retrograde transport and the central effect of botulinum toxin have also been indicated by earlier animal studies. Central antinociceptive activity of BoNT/A was shown in a rat model of trigeminal neuropathy, with BoNT/A-truncated SNAP-25 found in the spinal trigeminal nucleus 3 days after peripheral treatment [14]. Antonucci et al. also demonstrated retrograde transport of BoNT/A by detecting BoNT/A-truncated SNAP-25 in synaptic terminals of the retina three days post injection of the toxin into the optic rectum of Sprague-Dawley rats [15].

A neuroplastic effect, possibly centrally mediated, has been suggested by the partial post-BoNT/A restoration of the impaired levels of urothelial and suburothelial muscarinic receptors in overactive bladders [16]. Additionally, secondary changes in the CNS have been proposed following treatment with BoNT/A in pain syndromes [17].

Based on the above, we conducted an animal experiment to study the hypothesis that bladder OnabotulinumtoxinA injection affects the expression of genes in the spinal cord (SC) and dorsal root ganglia (DRG), which control bladder function. In normal rats, we explored possible changes of sensory, cholinergic and sympathetic markers at the L6-S1 SC segment and the respective DRG—which are known to be associated with the bladder afferent pathways in rats—following injection of the toxin into the animal bladder, and in parallel with respective bladder changes.

## 2. Results

### 2.1. Real-Time PCR Gene Expression Results

Sensory markers: Tac1 expression significantly decreased following treatment with 2U or 5U OnabotulinumtoxinA at 14 days but was not significantly altered at 7 days (Figure 1).

No significant changes were seen in TRPV1 bladder expression. TRPV1 expression in the DRG increased following the 5U injection at 7 days but returned to control levels at 14 days. Spinal TRPV1 expression was also increased after both the 2U and 5U injection at 7 days (Figure 1).

Cholinergic markers: Expression of muscarinic receptors M1 and M2 was decreased in the bladder and spinal cord, especially after the administration of 5U OnabotulinumtoxinA (Figure 2). Interestingly, expression of M1 and M2 in the DRGs was undetectable (Figure 2). No statistically significant changes were observed in the expression of M3 in all tissues.

Sympathetic markers: NPY expression increased in the DRGs both at 7 and 14 days in animals treated with 5U OnabotulinumtoxinA, as well as in animals injected with 2U OnabotulinumtoxinA at 7 days post injection. At 14 days post injection with 2U and 5U OnabotulinumtoxinA, a significant increase was noted in the SC. Bladder NPY expression showed a similar incremental trend, but only at 7 days with the 5U dose (Figure 3).

Proenkefalin (PENK): Expression of proenkefalin increased at 7 days post administration of 2U OnabotulinumtoxinA in the DRG and 5U OnabotulinumtoxinA in the SC (Figure 4).

All statistically significant *p*-values are presented in Table 1.

### 2.2. Protein Expression Results

For all genes with significantly affected expression as detected by real-time PCR, their respective protein levels were measured by Western blot analysis. Due to limited protein extracts from tissues, proteins could not be detected in all cases. We present quantified WB results for the proteins that were detectable (Figure 5).

Sensory markers: In the DRG, TRPV1 protein expression was not significantly affected after 2U, or 5U, OnabotulinumtoxinA compared with saline.

Cholinergic markers: Expression of muscarinic M1 receptor appeared to be decreased in the bladder after the administration of 2U and 5U OnabotulinumtoxinA at 7 and 14 days. M2 protein expression showed no change in the bladder, but it decreased in the SC after 14 days of treatment with 5U OnabotulinumtoxinA.

Sympathetic markers: NPY expression decreased in the DRG at 7 days in animals treated with either 2U or 5U OnabotulinumtoxinA.

PENK: Expression of proenkefalin increased at 7 days post administration of 5U OnabotulinumtoxinA in the DRG. There were no significant changes in the SC.

All statistically significant p-values are presented in Table 2.

### 2.3. Histology

Bladder: Full-thickness bladder specimens were examined. Minimal perivascular inflammatory infiltration was observed in the bladder of saline-treated animals, while in OnabotulinumtoxinA-treated animals (2U and 5U OnabotulinumtoxinA, irrespective of the time of death,), mild inflammatory infiltration (lymphocytes and plasma cells) was observed in the submucosa, which was focally extended to the serosa in 2 animals (Figure 6).

Spinal cord and DRGs: No histopathological lesions were detected in the saline-treated animals. In the OnabotulinumtoxinA-treated animals (2U and 5U OnabotulinumtoxinA, irrespective of the time of euthanasia), increased numbers of satellite cells and chromatolysis in a limited number of neurons were observed. Additionally, mild peri-ganglial infiltration by lymphocytes and plasma cells was observed in one animal (Figure 6).

## 3. Discussion

Our findings in normal rats are indicative of a central effect of bladder BoNT/A injection as it was followed by significant changes in the expression of sensory, sympathetic and cholinergic markers in ganglia and spinal cord segments, which are involved in the control of bladder function. All markers of gene expression demonstrated a central induction, while changes in protein levels did not always follow the gene expression changes measured by real-time PCR.

The endogenous tachykinins, substance P and neurokinin A, along with the peptides neuropeptide K and neuropeptide gamma, are encoded by the Tac1 gene [18]. Substance-P is a neuropeptide participating in the regulation of the lower urinary tract [19]. We found Tac1 gene expression significantly decreased, both in the bladder and SC, following treatment with 2U or 5U at 14 days. In DRG, a significant decrease was only noted with 2U at 14 days. In a similar study investigating the neurological characterization of DRG neurons supplying the porcine urinary bladder after retrograde tracing with FastBlue [FB], bladder injection of OnabotulinumtoxinA led to a significant decrease in the number of sensory neurons containing Substance-P (SP) [20]. In contrast, a central upregulation of SP in the raphe nucleus was reported after injection of botulinum toxin in twenty muscle groups in rats [21]. Inhibition of the release of SP as well as of other sensory neuropeptides, such as calcitonin gene-related peptide (CGRP), in the bladder was also found in earlier studies of isolated bladder preparations from acute injury or chronic inflammation bladder rat models after OnabotulinumtoxinA injection [22].

TRPV1 (vanilloid receptor) is involved in the sensation of bladder distention and fullness (mechano-sensation) and is abnormally expressed under conditions of detrusor overactivity (DO) [23], but also bladder pain [24]. Our experiment showed no significant changes in TRPV1 bladder expression, apparently contradicting findings from previous studies. Apostolidis et al. found that OnabotulinumtoxinA intradetrusor injection in patients with DO were associated with a reduction in suburothelial levels of the receptors TRPV1 (vanilloid receptor) and P2X_3_ (purinergic ATP-gated receptor) [23]. However, our experimental model involved normal rats as opposed to bladder pathology in previous studies [23]. In addition, our results came from whole bladder preparations (urothelium, suburothelium and detrusor muscle). Despite the lack of a bladder TRPV1 effect, we found increases in TRPV1 gene expression in both the DRG and the spinal cord. These changes were more immediate (7 days after the 5U injection in the DRG and after both the 2U and 5U injection in the SC) but returned to control levels at 14 days. Interestingly, protein expression level of TRPV1 was not significantly changed in the DRG and was not detected in the spinal cord under our experimental conditions. These data suggest a complex effect of BoNT/A on TRPV1 mRNA and protein expression, which may also reflect the available literature. Earlier in vitro experiments demonstrated a blockade of the SNARE-dependent TRPV1 exocytosis to the plasma membrane, suggesting a direct effect on receptor trafficking [25]. It was found that blocking TRPV1 trafficking to the plasma membrane markedly decreased total TRPV1 protein expression through proteasomal degradation [26,27]. In the latter study, subcutaneous BoNT/A injection in the face targeted the ophthalmic division of trigeminal ganglion (TG) neurons and decreased TRPV1-immunoreactive neurons in the TG and TRPV1-immunoreactive fibers in rat trigeminal terminals. However, quantitative real-time RT-PCR data indicated that the TRPV1-decreasing effects of BoNT/A were not mediated by transcriptional downregulation, since RT-PCR showed a slight increase in gene expression [27]. Another study of bladder biopsies from patients with neurogenic or idiopathic DO who underwent OnabotulinumtoxinA^−^intradetrusor treatment showed that the neurotoxin significantly increased TRPV1 gene expression independent of the etiology of DO [28].

We found similarly conflicting results regarding the expression of NPY, a sympathetic nervous system co-transmitter involved in the control of the lower urinary tract [19]. We noted a significant increase of NPY mRNA levels in bladder and DRG tissues but a decrease in SC in its protein levels in the DRG. Studies of intravesical BoNT/A injection in the pig have reported a decrease in the number of NPY-containing neurons in the bladder one week after injection [29], and an increase in the number of adrenergic NPY positive nerve terminals in the muscle layer of the bladder [30].

We observed downregulation in the expression of M1 and M2 receptors in both spinal cord and bladder samples obtained from normal rats after BoNT/A injections. Two studies in humans provide somewhat contrasting results, but this may be possibly attributed to the different bladder layers examined. Datta et al. reported an increase in immunoreactivity of M1 and M2 receptors in the suburothelium of DO patients treated with BoNT/A [31]. Schulte-Baukloh et al., on the other hand, ref. [5] reported a reduction in detrusor M2 and M3 receptors after bladder BoNT/A injections in children and adolescents with neurogenic DO, which was regarded as a contributing factor to the amelioration of detrusor overactivity and pressures.

Enkephalins are pentapeptides which comprise one of the three main groups of endogenous opioid peptides. Both enkephalins (met-enkephalin and leu-enkephalin) are produced via proteolytic cleavage from the precursor-polypeptide proenkephalin [32]. Several studies have confirmed the localization of enkephalins on the afferent pathways of the bladder and urethra [33], whereas enkephalinergic mechanisms in the brain and spinal cord have been proposed to inhibit the micturition reflex [18].

We observed an increase in proenkephalin expression 7 days post-BoNT/A administration both in the DRG and SC. Protein expression also increased 7 days post-BoNT/A in the DRG although there were no significant changes in the SC. These results suggest augmented central inhibition of bladder function and, if confirmed in overactive bladder models, could partly explain the inhibitory effect of BoNT/A on bladder overactivity. Interestingly, spinal cord ENK was one of the first genes which were found to be affected in earlier studies following BoNT/A injection in peripheral muscles in experimental animal models [10,11]. More specifically, injection of OnabotulinumtoxinA in the gastrocnemius muscle of rats was followed by upregulation of enkephalin and the acidic fibroblast growth factor (aFGF) in the lumbosacral spinal cord within 7–14 days post-injection [11].

We are aware of some discrepancies between the real-time PCR and Western blot results; however, this is a common occurrence which may be attributed to a number of reasons, such as differences in post-translational modifications between tissues. Furthermore, it is widely accepted that mRNA levels and protein levels are quite often regulated by different mechanisms. Contrary to expectations, a higher dose of OnabotulinumtoxinA (5U) did not always evoke changes in the expression of the studied genes although the lower dose did, but this inconsistency may be due to the small sample size of our study.

In addition to the gene expression changes both at the CNS level and in the urinary bladder, we obtained notable results from our preliminary histological study, which revealed alterations at the neuronal level coupled with inflammatory alterations in dorsal root neurons and ganglia. A previous study had shown a trend for augmentation of inflammatory alterations in the bladder wall after a single, but particularly after repeat, OnabotulinumtoxinA injections [34]. In general, reports concerning inflammatory histological findings after OnabotulinumtoxinA injection in the bladder and other tissues, both in human and animal models, appear to be contradictory [35,36,37]. In contrast, most reports agree on the lack of degenerative alterations on neural terminals [2,38]. Our finding of neuronal chromatolysis must therefore be further investigated since it implies a possible toxin effect. However, as our histological findings are based on examination of a single specimen per group of animals, these results can only be considered preliminary and indicative of the need for further investigations.

Our data using low doses of BoNT/A (2U and 5U) to inject the bladder of normal rats demonstrate a number of post-injection changes in cholinergic, sympathetic and sensory markers in the L6-S1 DRG and SC segment, which are known to be related to the control of lower urinary tract function. The mechanism for such a central effect remains to be elucidated. Two recent studies support the axonal transport of the toxin: intrathecal administration of BoNT/A resulted in the binding of SNAP25 on which ΒοΝΤ/A acts on, both in the spinal cord and in bladder nerve fibers [39]. Moreover, we recently reported a significant accumulation of radiolabeled BoNT/A in the lumbosacral DRG, together with an uptake in the respective SC segment following bladder injection in normal rats, providing the first evidence of BoNT/A’s retrograde transport to the CNS [13].

Limitations to our study are the lack of functional experiments, and the lack of prior data for power calculations in order to more accurately determine the appropriate sample size.

## 4. Materials & Methods

### 4.1. Animal Experiments

All animal experiments were performed in accordance with the guidelines set by the Aristotle University Committee for Animal Experimentation. Eight-week-old female Wistar rats were purchased (Animal Facilities of “Theageneio” General Hospital, EU License N^o^ LE54BIO31 & LE54BIO 32, Thessaloniki, Greece; 150–180 g) and housed in proper animal facilities (Laboratory of General Biology, Aristotle University, EU License N^o^ L54BIO36) for one week for veterinary–hygiene control with food and water *ad libitum* in constant conditions of temperature and humidity, and regular light cycles of 12/12 h light/dark. The rats were allocated to one of three groups: group I (n = 12) were administered 2U/50 μL BOTOX^®^, Allergan; group II (n = 12) were administered 5U/50 μL BOTOX^®^, Allergan; and group III (n = 12) were administered 50 μL vehicle. Briefly, the rats were sedated and anaesthetized with xylazine 5 mg/kg intraperitoneally (i.p.) (Rompun 2%, Bayer, Leverkusen, Germany) followed by ketamine 15 mg/kg i.p. (Imalgen 1000, Rhone, Merieux, Lyon, France). A 2-cm abdominal, midline, longitudinal incision was performed 5 mm above the external urethral meatus. The bladder was identified and equal doses of 10 μL were injected in the anterior, posterior, right and left lateral bladder wall, as well as in the bladder dome, employing an insulin syringe with a 28-gauge needle. The incision was closed with Vicryl 4.0 absorbable suture. One week after the injection, six animals were randomly selected from each group (subgroups Ia to IIIa) and killed in a chamber with continuous carbon dioxide flow. The bladder, the L6-S1 DRGs bilaterally, and the L6-S1 segment of the SC were isolated from each rat (Figure 7).

Upper panel: Left—surgical preparation of the urinary bladder in anaesthetized rat. Middle—injection of OnabotulinumtoxinA into the posterior bladder wall. Right—injection of OnabotulinumtoxinA into the anterior bladder wall.

Middle panel: Left—sacrifice of experimental animal by CO_2_. Middle—surgical preparation of the urinary bladder. Right—surgical preparation of the lumbosacral spinal region with identification of the pelvic nerve, the posterior and anterior L6-S1 roots, and the respective dorsal root ganglion

Lower panel: Left—isolated preparation of the L6-S1 spinal region. Middle—removal of the anterior spinal surface to reveal the spinal cord. Right—final preparation of the L6-S1 spinal cord, L6-S1 roots, and dorsal root ganglia

Tissue samples obtained from one animal were maintained in formaldehyde-based fixation solution for histological evaluation and samples from all three tissue types from the remaining animals were cryopreserved in liquid nitrogen for further evaluation with molecular biology techniques. Two weeks after the injection, the remaining 6 animals from each group (subgroups Ib to IIIb) were treated as described above.

### 4.2. Real-Time Polymerase Chain Reaction

Tissue samples were snap frozen in liquid nitrogen and stored at −80 °C until extraction. Total RNA was extracted with TRIzol reagent (Life Technologies) according to manufacturer’s instructions and subsequently quantified using NanoDrop™ Spectrophotometry (Thermo Scientific Inc., Waltham, MA, USA). Total RNA (500 ng) was DNase treated with the Ambion^®^TURBO DNA-free™ kit (Life Technologies) according to manufacturer’s instructions. DNase-treated RNA was reverse-transcribed using the SuperScript™ First-Strand Synthesis System (Invitrogen, Life Technologies Corporation). The relative expression was assessed using TaqMan^®^ Gene Expression Assays (Table 3). Real-time qPCR reactions were carried out in a StepOnePlus™ Real-Time PCR system (Applied Biosystems^®^, Life Technologies Corporation). Gene expression was assessed using the 2^−ΔΔCt^ method and presented as RQ values. Glyceraldehyde 3-phosphate dehydrogenase (GAPDH) was used as a reference gene and the calibrator sample consisted of equal amounts of five bladder control RNA samples.

### 4.3. Protein Extraction—Immunoblotting

Bladder tissue was lysed in RIPA buffer (containing 50 mM Tris-HCl (pH 8,0), 1% NP-40, 0.5% sodium deoxycholate, 0.1% SDS, 500 mM NaCl, 10 mM MgCl_2_ and a protease inhibitor cocktail (Sigma, P8340)), homogenized and incubated at 4 °C for 2 h. Insoluble material was removed by centrifugation at 12.000× *g* for 10 min at 4 °C. Protein concentration was determined using a commercial Bradford reagent (Bio-Rad protein assay kit, Hercules, CA, USA), and the samples were stored at −70 °C until analysis. Total protein extracts were analyzed by SDS-PAGE/Western blotting. Proteins were resolved on 10% SDS-polyacrylamide gels and electrotransferred onto nitrocellulose membranes. For immunodetection, goat polyclonal anti-TRPV1 (sc-12498, Santa Cruz), rabbit polyclonal anti-NPY (sc-14728-r, Santa Cruz), goat polyclonal anti-Substance-P (sc-14104, Santa Cruz), mouse monoclonal anti-met/leu Enkephalin (sc-47705, Santa Cruz), rabbit polyclonal anti-mAChRM1 (sc-9106, Santa Cruz), rabbit polyclonal anti-mAChRM2 (sc-9107, Santa Cruz), or mouse monoclonal anti-actin (clone C4/MAB150, Millipore, 1:1000 dilution) antibodies were used. Goat anti-mouse IgG coupled to horseradish peroxidase (AP130P, Millipore, 1:10,000 dilution), goat anti-rabbit IgG and donkey anti-goat IgG (Santa Cruz Biotechnology, Inc., 1:10,000 dilution) coupled to horseradish peroxidase were used as secondary antibodies. The enhanced chemiluminescence detection system was applied according to manufacturer’s protocol (GE Healthcare, RPN2209, Buckinghamshire, UK). Protein band intensity was quantified using PC-based image analysis (ImageJ, Java based image processing, NIH). The density of all bands was normalized to a reference sample loaded onto all gels and measured relative to actin-band density.

### 4.4. Histology

All specimens were routinely processed, paraffin embedded, sectioned at 5 μm, stained with hematoxylin and eosin (H&E) and then evaluated microscopically.

### 4.5. Statistical Analysis

Mean values were compared between groups with analysis of variance (ANOVA) when the data followed normal distribution, and with the non-parametric Kruskal-Wallis test when the data did not follow normal distribution. Groups in pairs were compared with the Bonferroni post-hoc analysis for normal distribution, and with the Mann–Whitney test for non-normal distributions. All values were expressed as mean value ± standard error, and differences with a *p* value of ≤0.05 were considered significant. All analyses were performed with the SPSS statistical package (version 21).

## 5. Conclusions

Our findings in normal rats suggest that bladder injections of OnabotulinumtoxinA may be followed by changes in the expression of sensory, sympathetic and cholinergic markers important in the regulation of bladder function at the DRG/SC level. Further experiments are deemed essential in order to confirm our results in bladder dysfunction states and to investigate central neuronal plasticity as a direct consequence of the toxin following its intradetrusor administration.

## Figures and Tables

**Figure 1 ijms-23-14419-f001:**
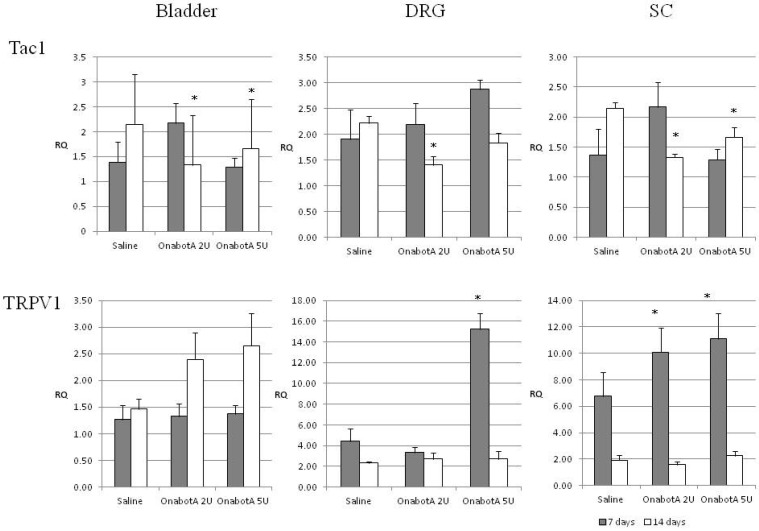
Relative expression of sensory markers Tac1 and TRPV1 in the bladder, DRG (dorsal root ganglia) and SC (spinal cord) of saline-treated (7 days post-treatment, n = 5; and 14 days post-treatment, n = 5) and OnabotulinumtoxinA-treated rats (7 days post-treatment, n = 5; and 14 days post-treatment, n = 5). Expression was normalized to that of GAPDH. RQ: relative quantification. Error bar: standard deviation. *: statistically significant difference (*p* < 0.05) with respect to saline.

**Figure 2 ijms-23-14419-f002:**
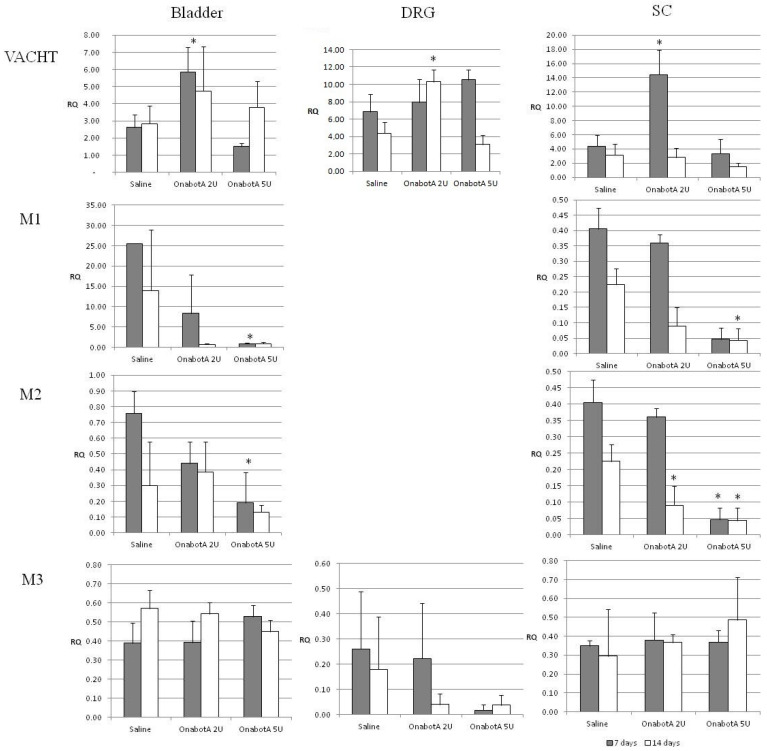
Relative expression of cholinergic markers M1, M2 and M3 in the bladder, DRG (dorsal root ganglia) and SC (spinal cord) of saline-treated (7 days post-treatment, n = 5; and 14 days post-treatment, n = 5) and OnabotulinumtoxinA-treated rats (7 days post-treatment, n = 5; and 14 days post-treatment, n = 5). Expression was normalized to that of GAPDH. RQ: relative quantification. Error bar: standard deviation. *: statistically significant difference (*p* < 0.05) with respect to saline.

**Figure 3 ijms-23-14419-f003:**
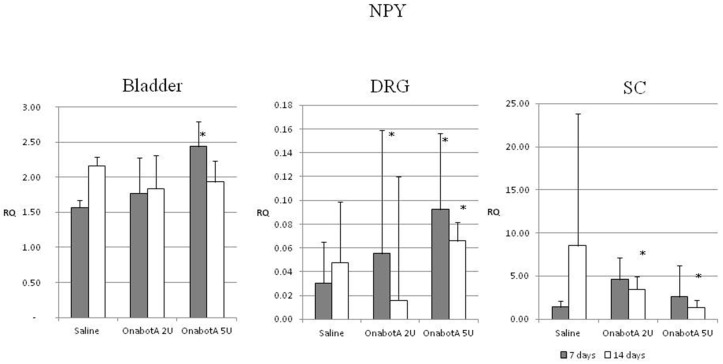
Relative expression of sympathetic marker NPY in the bladder, DRG (dorsal root ganglia) and SC (spinal cord) of saline-treated (7 days post-treatment, n = 5; and 14 days post-treatment, n = 5) and OnabotulinumtoxinA-treated rats (7 days post-treatment, n = 5; and 14 days post-treatment, n = 5). Expression was normalized to that of GAPDH. RQ: relative quantification. Error bar: standard deviation. *: statistically significant difference (*p* < 0.05) with respect to saline.

**Figure 4 ijms-23-14419-f004:**
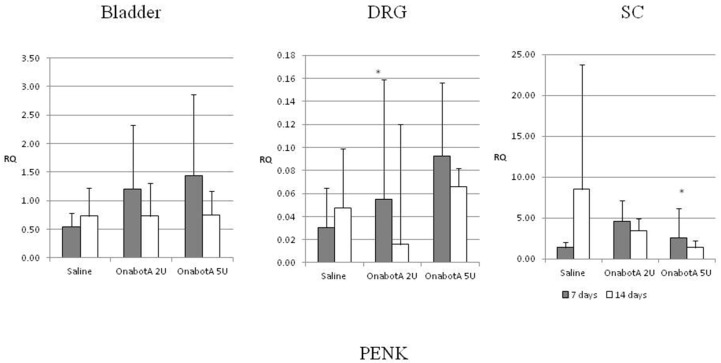
Relative expression of PENK in the bladder, DRG (dorsal root ganglia) and SC (spinal cord) of saline-treated (7 days post-treatment, n = 5; and 14 days post-treatment, n = 5) and OnabotulinumtoxinA-treated rats (7 days post-treatment, n = 5; and 14 days post-treatment, n = 5). Expression was normalized to that of GAPDH. RQ: relative quantification. Error bar: standard deviation. *: statistically significant difference (*p* < 0.05) with respect to saline.

**Figure 5 ijms-23-14419-f005:**
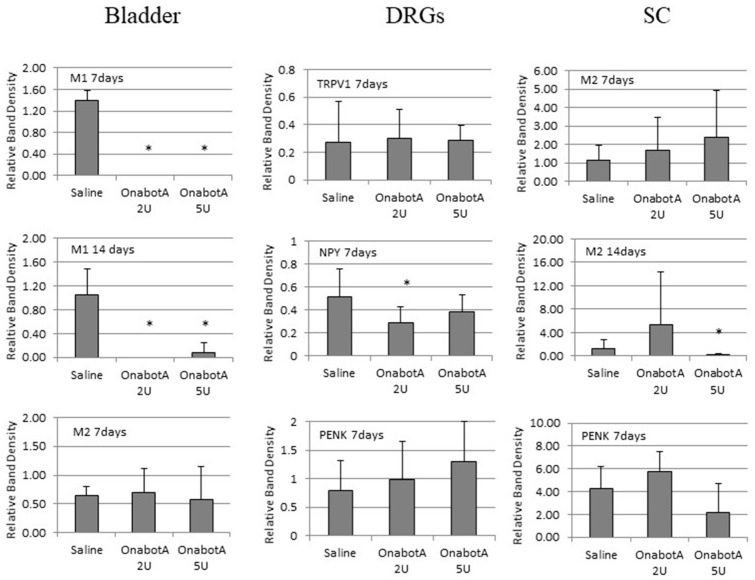
Quantified relative band density of proteins in the bladder, DRG (dorsal root ganglia) and SC (spinal cord) of saline-treated (7 days post-treatment, n = 5; and/or 14 days post-treatment, n = 5) and OnabotulinumtoxinA-treated rats (7 days post-treatment, n = 5; and/or 14 days post-treatment, n = 5). Band densities were normalized to a reference sample loaded onto all gels and measured relative to actin band density. Error bar: standard deviation. *: statistically significant difference (*p* < 0.05) with respect to saline.

**Figure 6 ijms-23-14419-f006:**
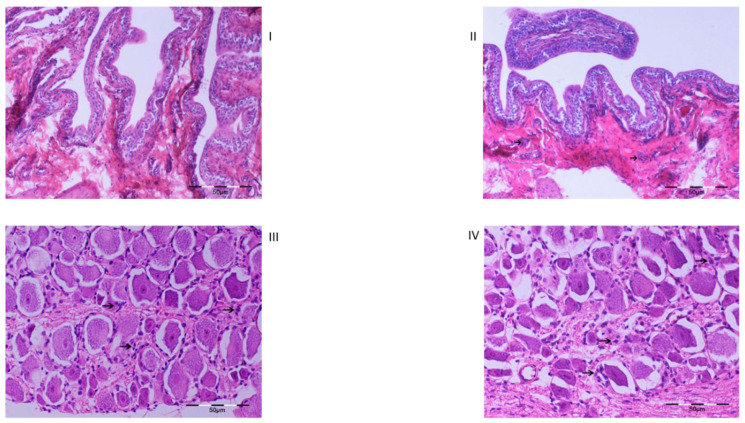
Images from histological examination of tissue sections of bladder and ganglia (hematoxylin and eosin staining, magnification ×200). Image **I**: no histopathological findings observed. Image **II**: mild infiltration from inflammatory cells is observed. Image **III**: no histopathological findings were observed. Image **IV**: small increase in the number of satellite cells is observed.

**Figure 7 ijms-23-14419-f007:**
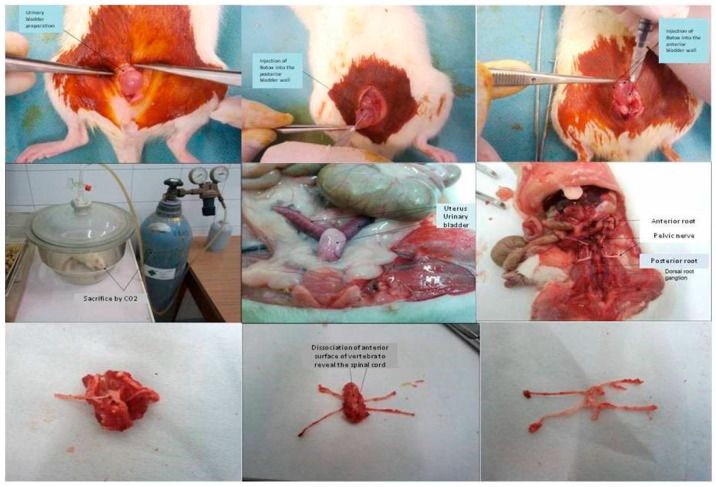
Representative photos from various stages of the feasibility experiment.

**Table 1 ijms-23-14419-t001:** *p*-values of statistically significant results of gene expression analysis by real-time PCR.

Gene	Treatment	*p*-Values		
		Bladder	Dorsal Root Ganglia	Spinal Cord
Tac1	$\begin{array}{ll} \mathrm{Saline}\;\mathrm{vs}.\;\mathrm{OnabotulinumtoxinA}\;2U & 14\mathrm{days}\\ \mathrm{Saline}\;\mathrm{vs}.\;\mathrm{OnabotulinumtoxinA}\;5U & \end{array}\}$	0.000 0.042	0.006 -	0.000 0.042
TRPV1	$\begin{array}{ll} \mathrm{Saline}\;\mathrm{vs}.\;\mathrm{OnabotulinumtoxinA}\;2U & 7\mathrm{days}\\ \mathrm{Saline}\;\mathrm{vs}.\;\mathrm{OnabotulinumtoxinA}\;5U & \end{array}\}$	- -	- 0.001	0.017 0.005
NPY	$\begin{array}{l} \begin{array}{ll} \mathrm{Saline}\;\mathrm{vs}.\;\mathrm{OnabotulinumtoxinA}\;2U & 7\mathrm{days}\\ \mathrm{Saline}\;\mathrm{vs}.\;\mathrm{OnabotulinumtoxinA}\;5U & \end{array}\}\\ \begin{array}{ll} \mathrm{Saline}\;\mathrm{vs}.\;\mathrm{OnabotulinumtoxinA}\;2U & 14\mathrm{days}\\ \mathrm{Saline}\;\mathrm{vs}.\;\mathrm{OnabotulinumtoxinA}\;5U & \end{array}\} \end{array}$	- 0.047 - -	0.028 0.028 - 0.028	- - 0.021 0.008
M1	$\begin{array}{ll} \mathrm{Saline}\;\mathrm{vs}.\;\mathrm{OnabotulinumtoxinA}\;5U & 7\mathrm{days}\\ \mathrm{Saline}\;\mathrm{vs}.\;\mathrm{OnabotulinumtoxinA}\;5U & 14\mathrm{days} \end{array}\}$	0.030 -	- -	- 0.014
M2	$\begin{array}{ll} \mathrm{Saline}\;\mathrm{vs}.\;\mathrm{OnabotulinumtoxinA}\;5U & 7\mathrm{days}\\ \mathrm{Saline}\;\mathrm{vs}.\;\mathrm{OnabotulinumtoxinA}\;2U & 14\mathrm{days}\\ \mathrm{Saline}\;\mathrm{vs}.\;\mathrm{OnabotulinumtoxinA}\;5U & \end{array}\}$	0.002 - -	- - -	0.000 0.052 0.003
PENK	$\begin{array}{ll} \mathrm{Saline}\;\mathrm{vs}.\;\mathrm{OnabotulinumtoxinA}\;2U & 7\mathrm{days}\\ \mathrm{Saline}\;\mathrm{vs}.\;\mathrm{OnabotulinumtoxinA}\;5U & \end{array}\}$	- -	- 0.065	0.047 -

**Table 2 ijms-23-14419-t002:** *p*-values of statistically significant results of quantified relative band density of protein analysis by Western-blot.

Protein	Treatment	*p*-Values		
		Bladder	Dorsal Root Ganglia	Spinal Cord
TRPV1	$\begin{array}{ll} \mathrm{Saline}\;\mathrm{vs}.\;\mathrm{OnabotulinumtoxinA}\;2U & 7\mathrm{days}\\ \mathrm{Saline}\;\mathrm{vs}.\;\mathrm{OnabotulinumtoxinA}\;5U & \end{array}\}$	- -	0.383 0.415	- -
NPY	$\begin{array}{ll} \mathrm{Saline}\;\mathrm{vs}.\;\mathrm{OnabotulinumtoxinA}\;2U & 7\mathrm{days} \end{array}$	-	0.010	-
M1	$\begin{array}{l} \begin{array}{ll} \mathrm{Saline}\;\mathrm{vs}.\;\mathrm{OnabotulinumtoxinA}\;2U & 7\mathrm{days}\\ \mathrm{Saline}\;\mathrm{vs}.\;\mathrm{OnabotulinumtoxinA}\;5U & \end{array}\}\\ \begin{array}{ll} \mathrm{Saline}\;\mathrm{vs}.\;\mathrm{OnabotulinumtoxinA}\;2U & 14\mathrm{days}\\ \mathrm{Saline}\;\mathrm{vs}.\;\mathrm{OnabotulinumtoxinA}\;5U & \end{array}\} \end{array}$	0.000 0.000 0.000 0.000	- - - -	- - - -
M2	$\begin{array}{ll} \mathrm{Saline}\;\mathrm{vs}.\;\mathrm{OnabotulinumtoxinA}\;5U & 14\mathrm{days} \end{array}$	-	-	0.050
PENK	$\begin{array}{ll} \mathrm{Saline}\;\mathrm{vs}.\;\mathrm{OnabotulinumtoxinA}\;2U & 7\mathrm{days}\\ \mathrm{Saline}\;\mathrm{vs}.\;\mathrm{OnabotulinumtoxinA}\;5U & \end{array}\}$	- -	- 0.054	- -

**Table 3 ijms-23-14419-t003:** TaqMan^®^ Gene Expression Assays and thermal cycling profile used for real-time PCR reactions.

Gene	Taqman Assay ID	Thermal Cycling Profile
PENK	Rn00567566	95 °C for 20 min, 45 cycles consisting of 95 °C for 1 min, 60 °C for 20 min.
TRPV1	Rn00583117	
NPY	Rn01410145	
Tac1	Rn01500392	
M1	Rn00589936	
M2	Rn02532311	
M3	Rn00560986	
GAPDH	Rn01775763	

## Data Availability

Data can be available by the authors upon request.

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
