# Peer review of "Effect of Bladder Injection of OnabotulinumtoxinA on the Central Expression of Genes Associated with the Control of the Lower Urinary Tract: A Study in Normal Rats"

_ijms, 2022, doi:10.3390/ijms232214419_

Round 1
Author Response
Reviewer 1
This project studies the potential mechanisms through which bladder injection of Botulinum toxin A (BoNT/A), exerts its effects in control of the lower urinary tract. Animal experiments with healthy female rat models were conducted to evaluate the expression of genes in the spinal cord (SC) and dorsal root ganglia (DRG) which control the function of bladder and the involvement of sensory, cholinergic and sympathetic systems, in parallel with respective bladder changes. Overall, this study is well-explained, and the authors have covered their bases very well with these experiments. There are several minor concerns as noted below:
Reply: Thank you for your positive outlook of our study and for your time dedicated to improve our manuscript with your comments. Please find below our replies to separate comments.
- It could be mentioned why standard deviation is so high in the data associated with the expression of PENK and NPY; this could influence the validity of the data.
Reply: This reasonably reflects the heterogeneity of responses among individuals. It doesn’t affect the validity but merely represents the range of values observed as a normally expected response to treatment.
- Lack of a bladder TRPV1 effect based the P value, does not correspond with the data shown Figure 2; since the same differences in other groups is considered as significant. How could this be explained?
Reply: The figure does appear a bit confusing with regard to the potential overlap between different groups on day 7. In order to confirm our findings, we repeated the statistical analysis for bladder TRPV1 and the p values for Anova were p= 0.99 while individual differences based on the Dunnett’s post-hoc test were for saline vs. 5U Btx = 0.436 and for saline vs. 2U Btx = 0.506
- Considering the study design of 7 and 14 days post-injection, why the expression of proenkefalin (PENK) has not been studied in other time points?
Reply: As mentioned in section 3.2. Protein expression results of the manuscript, western blot analysis was performed only for those proteins whose genes showed significant expression changes post-treatment in the real-time PCR analysis.
- Line 340 – it should be “on the other hand”
Reply: Corrected
- Line 303– In order to avoid confusion, this sentence better be started with the name of the research group (Apostolidis et al.) whose results are contradictory to the findings in this project.
Reply: Corrected
- Line 75-77 – This sentence could be re-written so that it’s more explicable to the readers.
Reply: The sentence has been rephrased as follows: Wiegand et al detected radioactivity in motoneurons after application of radioiodine labeled botulinum toxin; the authors explained this finding by proposing a retrograde transport of the toxin.
- The style of Error bars in the figures could be more consistent (Figure 3 and 6).
Reply: Thank you for noticing this, we have revised Figures 3 and 6 so that the error bars are now more consistent.
- Are there any studies in literature indicating the changes of M1 and M2 expression levels in DRG in different models? If so, it would be worth mentioning in the discussion.
Reply: Thank you for your comment. Unfortunately, we could not find studies describing changes in the DRG expression of M1 and M2 receptors in other animal models.
Reviewer 2 Report
Thank you for the opportunity to review the manuscript entitled, "Effect of bladder injection of OnabotulinumtoxinA on the central expression of genes associated with the control of the lower urinary tract: a study in normal rats." The topic is important. However, I have several comments to improve the quality of the manuscript.
1. In line 43 – do the authors mean randomized controlled studies?
2. Could the authors comment on why there were 12 rats in each group? Moreover, what is the rationale behind choosing six animals from each group after a week?
3. The authors mention that if data was not normally distributed, they performed the Kruskal-Wallis test. When was this the case? Could the authors provide histograms? Moreover, when this is the case, it is better to report medians (IQR), rather than mean +/- standard error.
4. Could the authors include a limitations section?
5. In table 2, what do the numbers 14 and 7 mean? Also please spell out DRG and SC. Could the authors highlight p values that were below 0.05?
Author Response
Reviewer 2
Thank you for the opportunity to review the manuscript entitled, "Effect of bladder injection of OnabotulinumtoxinA on the central expression of genes associated with the control of the lower urinary tract: a study in normal rats." The topic is important. However, I have several comments to improve the quality of the manuscript.
Reply: Thank you for your positive outlook of our study and for your time dedicated to improve our manuscript with your comments. Please find below our replies to separate comments.
- In line 43 – do the authors mean randomized controlled studies?
Reply: Yes, we have corrected this
- Could the authors comment on why there were 12 rats in each group? Moreover, what is the rationale behind choosing six animals from each group after a week?
Reply: We didn’t have previous data for power calculations therefore we used the standard design of groups of six animals. We performed post –hoc power calculations (https://clincalc.com/), which demonstrated a power of 71-82% for the different genes studied in this experiment.
- The authors mention that if data was not normally distributed, they performed the Kruskal-Wallis test. When was this the case? Could the authors provide histograms? Moreover, when this is the case, it is better to report medians (IQR), rather than mean +/- standard error.
Reply: This was the case for bladder NPY. Please find the respective Histogram below.
We would opt to keep it as part of the original Figure (as per the original submission) instead of adding a separate Histogram to the manuscript, unless the reviewer suggests otherwise.
- Could the authors include a limitations section?
Reply: The most significant limitations of our study are the lack of functional experiments and the lack of data for power calculations in order to determine sample size. We have included a sentence at the end of the Discussion section to highlight this.
- In table 2, what do the numbers 14 and 7 mean? Also please spell out DRG and SC. Could the authors highlight p values that were below 0.05?
Reply: DONE
